# Insulin-like Growth Factor-1 Prevents Hypoxia/Reoxygenation-Induced White Matter Injury in Sickle Cell Mice

**DOI:** 10.3390/biomedicines11030692

**Published:** 2023-02-24

**Authors:** Rimi Hazra, Holland Hubert, Lynda Little-Ihrig, Samit Ghosh, Solomon Ofori-Acquah, Xiaoming Hu, Enrico M Novelli

**Affiliations:** 1Pittsburgh Heart Lung and Blood Vascular Medicine Institute, Department of Medicine, University of Pittsburgh, Pittsburgh, PA 15260, USA; 2Department of Neurology, University of Pittsburgh, Pittsburgh, PA 15260, USA; 3Geriatric Research, Education and Clinical Center, Veterans Affairs Pittsburgh Health Care System, Pittsburgh, PA 15240, USA

**Keywords:** sickle cell disease, hypoxia inducible factor-1α, insulin-like growth factor 1, white matter injury, hypoxia, stroke

## Abstract

Occlusion of cerebral blood vessels causes acute cerebral hypoxia—an important trigger of ischemic white matter injury and stroke in sickle cell disease (SCD). While chronic hypoxia triggers compensatory neuroprotection via insulin-like growth factor-1 (IGF-1) and hypoxia inducible factor-1α (HIF-1α), severe bouts of acute hypoxia and subsequent restoration of blood flow (hypoxia/reoxygenation, H/R) overwhelm compensatory mechanisms and cause neuroaxonal damage–identified as white matter lesions–in the brain. The neuroprotective role of IGF-1 in the pathogenesis of white matter injury in SCD has not been investigated; however, it is known that systemic IGF-1 is reduced in individuals with SCD. We hypothesized that IGF-1 supplementation may prevent H/R-induced white matter injury in SCD. Transgenic sickle mice homozygous for human hemoglobin S and exposed to H/R developed white matter injury identified by elevated expression of non-phosphorylated neurofilament H (SMI32) with a concomitant decrease in myelin basic protein (MBP) resulting in an increased SMI32/MBP ratio. H/R-challenge also lowered plasma and brain IGF-1 expression. Human recombinant IGF-1 prophylaxis significantly induced HIF-1α and averted H/R-induced white matter injury in the sickle mice compared to vehicle-treated mice. The expression of the IGF-1 binding proteins IGFBP-1 and IGFBP-3 was elevated in the IGF-1-treated brain tissue indicating their potential role in mediating neuroprotective HIF-1α signaling. This study provides proof-of-concept for IGF-1-mediated neuroprotection in SCD.

## 1. Introduction

Sickle cell disease (SCD) affects 1 in 1000 individuals worldwide [1,2] and is caused by mutated hemoglobin (hemoglobin S, HbS) that polymerizes when deoxygenated leading to the deformation of red blood cells into sickle cells, a process that causes their premature destruction (hemolysis). Hemolysis and its downstream effects cause a chronic inflammatory state characterized by the overexpression of adhesion molecules in the endothelium and blood cells. Hyperadhesive interactions between blood cells and the endothelium in turn cause vaso-occlusion which results in hypoxia, ischemia and infarction [3]. Vaso-occlusion-induced hypoxia and reoxygenation from the restoration of blood flow (hypoxia/reoxygenation, H/R) cause a number of downstream pathogenic effects including high oxidative stress and endothelial inflammation [3,4]. Thus, hypoxia is both a trigger and a consequence of vaso-occlusion in SCD. 

In the brain, H/R causes a wide range of complications including overt stroke and silent cerebral infarction [5,6,7]. About 40% of children with SCD develop infarcts, and the incidence of stroke increases by ~25% by the time they reach adulthood [8,9]. Stroke in SCD primarily affects the white matter regions of the brain because of their vulnerability to hypoxia, particularly in watershed areas that are not perfused by collaterals [10,11]. Pathological findings in these areas include intimal hyperplasia and stenosis of the internal carotid artery, intraluminal thrombosis, and infarcted cortical and frontoparietal deep white matter border areas with neuroaxonal damage [12,13,14]. 

Hypoxia inducible factor-1α (HIF-1α) plays a key role in maintaining oxygen homeostasis. HIF-1α serves as an oxygen-dependent transcriptional activator that is upregulated during hypoxia and acts as a master regulator for hypoxia-responsive genes that provide neuroprotection following cerebral ischemia [15]. Following activation, HIF-1α regulates the transcription and translation of several target genes that are involved in angiogenesis, glucose transport and metabolism, and cell survival following cerebral ischemia [16]. The combined effect of these adaptive mechanisms is a neuroprotective role mediated by neurogenesis and cell regeneration in the aftermath of ischemic stress [17]; there is ample evidence that in the murine middle cerebral artery occlusion model of stroke increased HIF-1α expression during focal ischemia attenuates brain damage [18,19]. 

Insulin-like growth factor-1 (IGF-1) plays a pivotal role in the development, plasticity and survival of neurons [20]. In the brain, both locally and peripherally secreted IGF-1 possesses both endocrine and paracrine/autocrine activities and regulates numerous cerebral functions including myelination, synaptogenesis, neurogenesis, neuroprotection and neurotransmission [21,22]. Several epidemiological studies including the Framingham Study have shown that low baseline circulating IGF-1 is associated with an increased risk of ischemic stroke [23,24]. Moreover, it has been shown that systemic injections of IGF-1 improve neuronal health in hypertensive rat models with ischemic stroke [25]. Among its many neuroprotective functions, IGF-1 stimulates the activity and expression of HIF-1α [26] and accelerates its nuclear translocation [27,28]. In addition, the IGF binding proteins (IGFBPs), including IGFBP1 and IGFBP3, enhance the stability of IGF-1 and increase the binding affinity of IGF-1 to its cellular receptors [29], thus potentiating its neuroprotective effects [30,31]; higher IGF-1 and IGFBP3 plasma levels have been associated with better neurofunctional outcomes following ischemic stroke [32]. Plasma IGF-1 levels are instead low in humans with SCD [33] and in sickle mice that express human hemoglobin S and phenocopy many of the complications of SCD [34,35], suggesting that IGF-1-mediated neuroprotection may be impaired in SCD. However, the role of IGF-1 in acute or silent stroke in SCD has never been explored. In this study, we probed the IGF-1-HIF-1α axis in sickle mice and tested the efficacy of exogenous IGF-1 supplementation in attenuating hypoxia-reperfusion (H/R)-injury, a known trigger of white matter cerebral infarction in transgenic sickle mice [36]. 

## 2. Materials and Methods

*Animals*: The knock in Townes’ sickle cell mice [37] expressing human HbS were purchased from The Jackson Laboratory (Strain# 013071). Mouse genotypes were confirmed either by PCR or Hb gel electrophoresis. All animal experiments were performed under approval by the Institutional Animal Care and Use Committee at the University of Pittsburgh (Protocol #22010095). All mice were 12–14 weeks old at the time of the experiments.

*Hypoxia-Reoxygenation (H/R) Challenge:* For H/R experiments, homozygous SS mice housed in their standard cages were transferred into a hypoxia chamber (BioSpherix) fitted with a ProOx 110 gas controller (BioSpherix) in line with a high-pressure double gauge mixed gas primary nitrogen regulator fitted onto a vaporized nitrogen source. Mice were exposed to 7% oxygen for 2 h (hypoxia) and returned to room air (reoxygenation) for 1 h. They were then phlebotomized by retro-orbital bleeding using a capillary tube internally coated with heparin/EDTA anticoagulant

*IGF-1 Treatment:* Human recombinant IGF-1, (50 µg/kg, ab9573) or vehicle (phosphate buffer saline, PBS) were injected subcutaneously for 5 days and mice were then subjected to H/R challenge following IGF-1 or vehicle treatment on day 5.

*Immunofluorescence:* The brains from the experimental mice were removed immediately at the end of each experiment and following euthanasia in a carbon dioxide chamber filled at 30–70% saturation level. The organs were then fixed in 10% buffered formalin overnight (~12 h) and then processed for histology using paraffin embedding followed by sectioning (4 μm) on a positively charged glass slide. The brain tissue sections were rehydrated using graded alcohol and purified distilled water for double immunofluorescence staining using antibodies for non-phosphorylated anti-neurofilament H (SMI32, Biolegend, San Diego, CA, USA, cat# 801702) and myelin basic protein (MBP, Proteintech, Rosemont, IL, USA, cat# 1048-1-AP), two markers of white matter injury. Following overnight incubation with primary antibodies, the slides were counter-probed with appropriate secondary antibodies (Alexa594 A11005 Invitrogen; Alexa488 A11034, Life Technologies, Gaithersburg, MD, USA) conjugated with fluorescence probes. The slides were mounted with ProLong gold antifade (P36934, Invitrogen, Carlsbad, CA) and images were captured using an Olympus AX70 microscope and CellSens software (Microscopy Technologies, Tokyo, Japan). The fluorescent densities for red and green fluorescence within the merged image were calculated using the NIH Image J software (1.53k; NIH, Bethesda, MD, USA). The relative staining intensity of SMI32 (red) and MBP (green) denotes the SMI32/MBP ratio, a marker of white matter injury under various experimental conditions.

*Isolation of brain tissue RNA and real-time PCR for gene expression*: 

RNA isolation. Total RNA was isolated from brain tissues by homogenizing each sample in Trizol (Invitrogen). The isolated RNA was then reverse transcribed using a cocktail containing 5 μL of 10 μL~ RT buffer, 10 mM dNTP mix, 10 μL~ random hexanucleotide and Multiscribe RT 5 U/μL and RNAase free water. The mixture was incubated in a thermal cycler at 25 °C for 10 min and then at 37 °C for 120 min. The resulting cDNA samples were stored at −20 °C. All reagents were obtained from Applied Biosystems, Waltham, MA, USA.

Quantitative PCR. Quantitative real-time PCR was performed using a 7900HT Fast instrument and the SDS 2.3 software (Applied Biosystems). A predeveloped primer and probe set (TaqMan assays; Applied Biosystems) was used to analyze mRNA levels of 18S (Mm04277571_s1), Igfbp1(Mm00515154_m1), and Igfbp3 (Mm01187817_m1). The reaction for each cell sample was performed in triplicate and using the 40 cycle thermal cycling program: cycle 1–20 s at 95 °C; cycles 2 through 95 °C for 1 s, followed by 60 °C for 20 s.

*ELISA assay:* Freshly collected, anticoagulated blood samples were centrifuged at 1300× *g* for 15 min to obtain plasma. The whole brain tissues were placed into liquid nitrogen for snap freezing and then homogenized in RIPA lysis buffer containing a cocktail of protease and phosphatase inhibitors (1X). The brain homogenate protein was then collected as a supernatant following centrifugation of the homogenized mixture at 14,000× *g* for 10 min at 4 °C. We measured the level of HIF-1α (R&D Systems, Minneapolis, MN, USA, DYC1935) in brain tissue homogenate and IGF-1 (Abcam, Boston, MA, USA, ab100695) in the plasma and the brain tissue homogenate from sickle mice by ELISA using manufacturer’s instructions. The total protein concentration was determined in the brain tissue homogenate to normalize the amount of HIF-1α and IGF-1 in the brain tissue.

*Statistical analysis:* The data were analyzed using a two-tailed unpaired Student’s *t*-test to determine the statistical significance between experimental groups. GraphPad Prism 9 software (Ver. 9.5.0; GraphPad Software, Boston, MA, USA) was used for generating the graphs and performing all statistical analyses. Data are presented as mean ± SEM. All *p*-values are reported in the figure legends. A *p*-value less than 0.05 was considered significant.

## 3. Results

We first set out to reproduce the acute cerebral ischemic damage observed in sickle mice in individuals with SCD after hypoxic triggers. We specifically probed whether H/R induces white matter lesions in mice, which in patients with SCD are evidence of loss of microstructural tissue integrity, axonal damage, loss of myelin, and disorganization within the cerebral white matter [11]. We found that following the H/R challenge, the expression of SMI32 was elevated and that of MBP was reduced, thereby significantly increasing the SMI32/MBP ratio, an indicator of white matter injury [38], as compared to mice subjected to normoxia (Figure 1A). The expression of HIF-1α was slightly elevated in the brain of mice subjected to H/R, suggesting the brain of sickle mice was undergoing an adaptive metabolic response to hypoxia (Figure 1B). However, evidence of white matter injury despite HIF-1α induction suggested that the increase in HIF-1α was insufficient to protect sickle mice brains from H/R-induced white matter damage. Indeed, we observed that expression of IGF-1 was significantly depleted both in the plasma as well as in the brain tissue following the H/R challenge (Figure 1C,D). 

We posited that relative IGF-1 deficiency may have hampered adequate production of HIF-1α, and tested whether this could be reversed by IGF-1 supplementation. Accordingly, we prophylaxed the sickle mice with human recombinant IGF-1 prior to the H/R challenge. We found that multiple injections of IGF-1 resulted in restored neuroaxonal integrity compared to vehicle-injected sickle mice following H/R, which was demonstrated by a significant reduction in the SMI32/MBP ratio in the IGF-1 treated group (Figure 2A). We also observed a significant increase in the total IGF-1 and HIF-1α contents in the IGF-1-treated sickle mouse brain tissue (Figure 2B,C). These data indicate that adding exogenous IGF-1 to the circulation may further induce tissue HIF-1α and rescue the sickle brain from H/R-induced white matter injury. Finally, we found that the mRNA expressions of IGFBP1 and IGFBP3 were increased by seven-fold and four-fold, respectively (Figure 2D), suggesting that stimulation of cerebral HIF-1α may be mediated by activation of IGFBPs.

## 4. Discussion

Stroke and silent cerebral infarction remain a common complication of SCD with many patients accruing white matter lesions detectable by magnetic resonance imaging over their lifetime [39]. A high burden of white matter lesions is associated with worse cognitive performance in children with SCD [40,41]. To date none of the existing disease-modifying strategies for SCD fully prevents neurovascular complications [42], partly because their pathogenesis is incompletely understood; a better understanding of the molecular mechanisms leading to brain injury is, therefore, needed to develop effective preventive and therapeutic strategies.

This is the first report of a beneficial role of IGF-1 prophylaxis in the prevention of white matter injuries in a preclinical model of SCD. We hypothesized that the lack of optimal induction of HIF-1α following H/R may trigger axonal damage within the white matter area in sickle mouse brains and that IGF-1 supplementation could afford neuroprotection in this model by leading to HIF-1α overexpression. Our hypothesis was prompted by the finding that patients with SCD and sickle mice have reduced circulating IGF-1 levels. We indeed found that IGF-1 prophylaxis reduced H/R-mediated white matter damage, although further studies will need to elucidate the mechanism of IGF-1-mediated HIF-1α overexpression in our model. Under normoxia, prolyl hydroxylase domain proteins (PHDs) regulate HIF-1α stability through ubiquitination and degradation of HIF-1α by the E3 ubiquitin-ligase von Hippel-Lindau protein (VHL) [43,44], and inhibition of PHDs induces HIF-1α activity [45,46]. It will be interesting to test whether elevated IGF-1 can inhibit PHDs to block the canonical pathway of HIF-1α ubiquitination and degradation, thereby accelerating brain HIF-1α expression in the sickle mouse brain.

Another potential neuroprotective mechanism of IGF-1 prophylaxis includes the induction of IGFBPs. IGF-1 and IGFBP3 are expressed by pericytes and astrocytes in the brain [47] and both IGF-1 and IGFBP3 mRNA and proteins increase in the brain of rat models following hypoxic injury thereby protecting from neuronal loss [48]. Additionally, IGFBP3 enhances the half-life of IGF-1 [49]. Its expression is elevated in cerebral microvascular endothelial cells following traumatic or hypoxic injury but decreased in neurons after hypoxia [50]. In vitro studies have shown that IGFBP3 induces cell growth through epidermal growth factor receptor phosphorylation in an IGF-1 independent manner [51]. In our study, we found significant induction of IGFBP3 mRNA following IGF-1 treatment, so supplementation of IGFBP3 may also be neuroprotective following H/R challenge in SCD.

In summary, we have provided preliminary evidence of a potentially new mechanism of neuroprotection in SCD. However, further studies are warranted to determine the efficacy of IGF-1 if given post-H/R challenge. In addition to the therapeutic potential of modulating the IGF-1-HIF-1α pathway, plasma levels of IGF-1 and IGFBP3 may represent molecular risk indicators for the development of white matter injury in SCD patients. Probing the IGF-1 system may, therefore, lead to a molecular biomarker-based precision medicine approach to identify patients who are both at risk of stroke and amenable to IGF-1 targeting interventions.

## Figures and Tables

**Figure 1 biomedicines-11-00692-f001:**
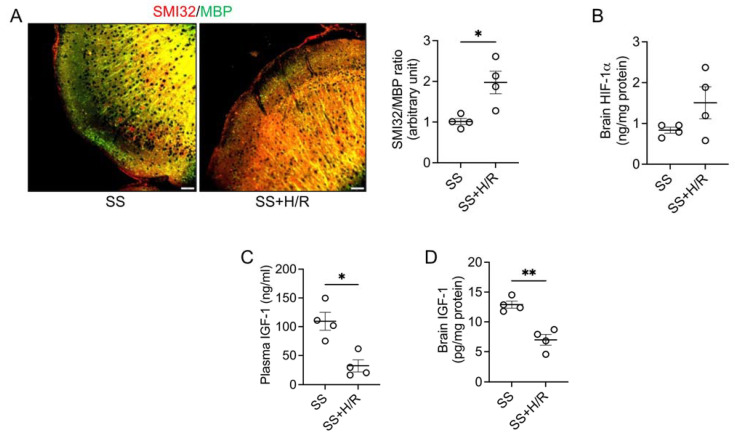
Hypoxia/Reoxygenation induced white matter injury is associated with reduced IGF-1 in sickle mice. (**A**) Sickle mice were either kept on room air (SS) or challenged with 7% oxygen for 2 h followed by 1 h of normoxia (SS+H/R). Representative immunofluorescence microscopic images showing merged SMI32 (red) and MBP (green) staining (scale bar = 50 μm). The ratio of staining intensity of SMI32 and MBP indicates white matter injury in the SS+H/R mice compared to SS mice (*n* = 4). (**B**) Total HIF-1α protein levels were measured in brain tissue homogenates from SS and SS+H/R mice. HIF-1α was induced following H/R challenge (*n* = 4). (**C**,**D**) Total IGF-1 levels in plasma (**C**) and brain tissue (**D**) were reduced in the SS mice after H/R (*n* = 4). * *p* < 0.05 and ** *p* < 0.01.

**Figure 2 biomedicines-11-00692-f002:**
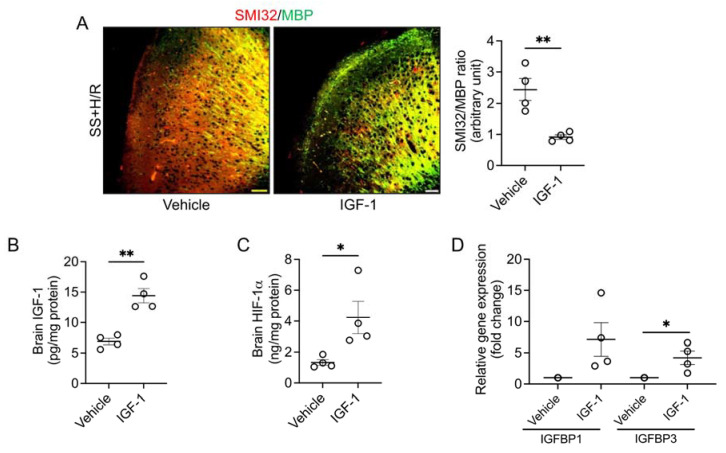
IGF-1 pretreatment protects sickle mice from H/R induced white matter injury. (**A**) The sickle mice (SS) were given subcutaneous injections of vehicle or human recombinant IGF-1 (50 µg/kg) for 5 days followed by H/R. Representative microscopic images of SMI32 (red) and MBP (green) double-stained brain sections collected from vehicle or IGF-1 treated SS mice (scale bar = 50 μm). The IGF-1 treated SS mice had reduced SMI32/MBP ratio indicating improved white matter integrity compared to vehicle-treated SS mice. (*n* = 4) (**B**) IGF-1 pretreatment increased IGF-1 content in the brains of SS mice. (*n* = 4) (**C**) HIF-1α concentration in brain tissue homogenates from vehicle and IGF-1 treated SS mice. (*n* = 4) (**D**) Relative expression of IGFBP1 and IGFBP3 genes isolated from brain tissues of vehicle or IGF-1 treated SS mice. (*n* = 4) * *p* < 0.05 and ** *p* < 0.01.

## Data Availability

All data are included in the manuscript.

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
