# Peer review of "Insulin-like Growth Factor-1 Prevents Hypoxia/Reoxygenation-Induced White Matter Injury in Sickle Cell Mice"

_biomedicines, 2023, doi:10.3390/biomedicines11030692_

Round 1
Reviewer 1 Report
This is a straightforward, well-defined, and hypothesis-driven manuscript. The authors provide excellent background information concerning mechanisms of white matter injury in sickle cell and a solid background for how IGF is modulated after stroke and neuro injury. The experiment is well-designed, and the data support their conclusions. The statistical analysis section needs to provide information concerning how the authors report their data. Are data reported as mean +/- SD or SEM? Data should be reported as mean +/- SD.
To confirm this study is composed of two experiments, where each experiment was performed with 4 animals in each test group. Although an n of 4 was used, some data appear to be n=3 or even 1. Is this because the data points overlap? If that is the case, can overlapping data points be offset, so all data points are visible? I recommend the authors consider subjecting more SS mice to H/R injury to increase the n for experiment 1B to determine if it can achieve statistical significance. Did they, by chance, do a power calculation with their current data to determine how many n would be required?
Author Response
Please see the attached file for the authors' response to the comments from Reviewer 1

Reviewer 2 Report
These authors have produced a nice but brief study showing that IGF-1 protects against hypoxic/reoxygenation injury in the brains of SCD mice. The study is generally well done and the results are clearly presented. The data generally support the conclusions. I have only a few minor comments that should be addressed:
1. The data shown in Figs 1 and 2 should be combined (as much as possible) so that each graph shows data for SS, SS+HR and Vehicle and IGF1. This would facilitate comparisons. Indeed, I wonder if both current figures contain the same SS+HR controls.
2. Please do not break figure legends between pages.
3. The authors should include in the discussion a statement that IGF1 treatment in human patients may face the additional issue that it may be less effective if delivered after an HR event has already occurred.
Author Response
Please see the attached authors' responses to the comments from Reviewer 2
